# Minigene Splice Assays Allow Pathogenicity Reclassification of *RPE65* Variants of Uncertain Significance

**DOI:** 10.3390/genes16091022

**Published:** 2025-08-28

**Authors:** Daan M. Panneman, Erica G. M. Boonen, Zelia Corradi, Frans P. M. Cremers, Susanne Roosing

**Affiliations:** 1Department of Human Genetics, Radboud University Medical Center, 6525 GA Nijmegen, The Netherlands; erica.boonen@radboudumc.nl (E.G.M.B.); zelia.corradi@radboudumc.nl (Z.C.); frans.cremers@radboudumc.nl (F.P.M.C.); susanne.roosing@radboudumc.nl (S.R.); 2The Rotterdam Eye Hospital, Rotterdam Ophthalmic Institute, 3011 BH Rotterdam, The Netherlands

**Keywords:** inherited retinal diseases, variant interpretation, ACMG classification, splice assays

## Abstract

Background/objectives: Obtaining a genetic diagnosis for patients with inherited retinal diseases has become even more important since gene-specific therapies have become available. When genetic screening reveals variants of uncertain significance (VUS), additional evidence is required to determine genetic eligibility for therapy. Confirming the effect on splicing that is predicted by SpliceAI could change their classification to either likely pathogenic or pathogenic and would therefore be of great importance when interpreting these variants when geneticists worldwide are trying to reach a diagnosis. Methods: Using minigene assays, we established a pipeline to assess the effect on splicing for all variants. We selected 73 *RPE65* variants that were classified as either VUS or likely benign in the *RPE65* Leiden Open Variant Database (LOVD) or ClinVar and were predicted to affect splicing by SpliceAI with a delta score of >0.1 and by using an analysis window of 5000 bp up- and downstream of the variant. Results: Using four wild-type vectors, we generated 59 constructs containing the variants of interest. Through these minigene assays, we assessed the effect on splicing of these VUS to enable reclassification. Upon quantification, we identified seven variants with a full, aberrant splicing effect without residual wild-type transcript. Eleven variants had between 5% and 20% remaining wild-type transcript. Forty-one variants had ≥20% residual wild-type transcript, among which fifteen variants showed no effect on splicing. Conclusions: Following the 2023 established ClinGen specific ACMG guidelines for *RPE65* (Criteria Specification Registry), evidence from splice assays enabled reclassification of seven *RPE65* variants from VUS to pathogenic through an assigned PVS1-very-strong criterium, as less than 5% of wild-type transcript was present. These findings contribute to the interpretation of variants observed in patients, which will in turn dictate their eligibility for gene therapy.

## 1. Introduction

The importance of obtaining a genetic diagnosis for patients with inherited retinal diseases (IRDs) has increased since gene-specific therapies have become available. Currently, the Luxturna *RPE65* gene augmentation therapy is being used to treat patients with Leber congenital amaurosis (LCA) and retinitis pigmentosa (RP) carrying *RPE65* variants [1,2]. When genetic screening reveals homozygous or compound heterozygous variants in *RPE65*, variants of uncertain significance (VUS) require additional evidence to determine whether patients are eligible for therapy. For VUS that are predicted to affect the splicing of mRNA transcripts, this additional evidence can be provided with in vitro splice assays. SpliceAI is used as an in silico prediction tool when predicting potential effects of variants on splicing, which has been proven to be a reliable predictor [3,4]. Alterations in splicing can lead to severe effects on mRNA and, ultimately, protein level. However, these predictions by themselves are not sufficient evidence to increase the ACMG classification from VUS to likely pathogenic or pathogenic, as they are still deemed to be supporting evidence (PP3-supporting) [5]. To do so, functional data, e.g., using an in vitro splice assay, needs to be added. As per the ClinGen specific ACMG guidelines for *RPE65* (Criteria Specification Registry (genome.network), accessed in September 2024) that were established in 2023, evidence from splice assays that confirm complete absence of wild-type transcripts can be assigned the PVS1-very-strong criterium. When less than 5% of wild-type transcript is present, PVS1-strong can be assigned [6].

Previously, we reported the synonymous c.675C>A variant in *RPE65* that was classified as VUS. Minigene analysis confirmed the predicted splice defect of complete skipping of exon 7 which, ultimately, led to the re-classification of the c.675C>A from VUS to pathogenic [7]. In this study, we selected 75 *RPE65* variants that were classified as either VUS, likely benign, or benign in the *RPE65* Leiden Open Variant Database (LOVD) or ClinVar and were predicted to affect splicing. Using minigene assays, we assessed the effect on splicing of these VUS to enable reclassification.

## 2. Materials and Methods

### 2.1. Variant Selection

All *RPE65* variants that were submitted to LOVD (www.lovd.nl/RPE65, accessed in September 2024) and ClinVar (https://www.ncbi.nlm.nih.gov/clinvar/?term=RPE65[, accessed in September 2024) were extracted in October 2023. Using Franklin by Genoox (franklin.genoox.com, accessed in September 2024), ACMG classifications were assigned to all variants. For variants with a classification of likely benign or VUS, SpliceAI was used to predict aberrant splicing. All SpliceAI scores of >0.1 on any of the four parameters (acceptor gain, acceptor loss, donor gain, donor loss) with a window of 5000 bp up- and downstream of the variant were taken forward. After removing duplicate variants from both the LOVD database and ClinVar, we prioritized 73 variants to test using the minigene analysis (Table 1).

### 2.2. Minigene Construct Generation

Wild-type minigene constructs were generated as previously described [8]. In short, we generated entry clones WT-1, WT-2, WT-3, and WT-4 spanning exons 1 to 5, exons 3 to 6, exons 6 to 10, and exons 11 to 13, respectively. The regions of interest were amplified by primers that contain attB1 and attB2 tags at their 5′ and 3′ end and were subsequently cloned into the pDONR201 vector using the Gateway cloning system. After obtaining the WT-1, WT-2, WT-3, and WT-4 entry clones, variants of interest were inserted individually using site-directed mutagenesis, as published previously [9]. Wild-type and mutant constructs were confirmed using Sanger sequencing. For variants that were not successfully introduced, the following steps were taken before excluding the variant: (1) performing mutagenesis using Phusion polymerase as previously described [9], (2) using Q5 high-fidelity polymerase (New England Biosciences), and (3) using a re-annealing protocol where forward and reverse primers were added to two separate reactions that were ultimately reannealed together using the following protocol: 5 min at 95 °C, 1 min at 90 °C, 1 min at 80 °C, 30 s at 70 °C, 30 s at 60 °C, 30 s 50 °C, and 30 s at 40 °C. Excluded variants are listed in Appendix A.

Next, mutant constructs containing the variants of interest and their corresponding wild-type construct were separately inserted into the *pCI-NEO-RHO* Gateway-adapted vector to generate wild-type and mutant minigenes. Both minigenes were independently transfected into HEK293T cells, and after 48 h of incubation, mRNA was isolated and amplified by RT-PCR with primers in the flanking *RHO* exon 3 and 5 regions. Primers used for these splice assays are listed in Appendix A. Fragment sizes were evaluated using gel electrophoresis and identified using Sanger sequencing. To quantify the ratios between correct and aberrant RT-PCR products, densitometric analysis was performed using Fiji software [10]. All constructs were transfected in duplicate in separate experiments and quantification and identification of transcripts was performed in two representative experiments. Gel images are available in Appendix A.

## 3. Results

We selected 73 *RPE65* variants from ClinVar and LOVD with a SpliceAI score of ≥0.1 in any of the four parameters and an ACMG classification of VUS or lower (Table 1). Using four wild-type vectors (WT-1 to 4), we generated 59 constructs containing the variants of interest. For 14 variants, we could not generate constructs, and these were therefore excluded from future experiments. All results of the minigene splice assays are provided in Table 2, and an overview is given in Figure 1.

Upon quantification, we identified seven variants with a full, aberrant splicing effect without residual wild-type transcript (Table 2). Eleven variants had between 5% and 20% remaining wild-type transcript. Forty-one variants had ≥20% residual wild-type transcript, among which 15 variants showed no effect on splicing. For the WT-3 construct encompassing exons 6–10, we observed two transcripts: a wild-type transcript (61%) and a transcript where exon 7 was skipped (39%). This is in line with previous observations and is considered a naturally occurring splicing event that is also observed in photoreceptor precursor cells and cultures of RPE-cells [7]. For WT-4, we observed, in addition to the wild-type transcript, a transcript in which exon 12 was skipped (17%) as well as a transcript with a 76-bp pseudo-exon containing *RPE65*: c.1129-293_1129-218 (6%).

Five out of the seven variants that had no residual wild-type transcripts were non-canonical splice site (NCSS) variants (c.643+5G>A, c.644–5T>A, c.999–3C>G, c.1338+3A>T, and c.1339–3C>G), and two variants were missense variants (c.650A>T and c.1334A>G). All variants resulted in protein truncation due to a frameshift. For eleven variants, we observed between 5% and 20% of wild-type transcript (c.294C>A, c.675C>T, c.676G>A, c.676G>T, c.701G>A, c.708G>A, c.713C>G, c.717C>T, c.726–3C>A, c.1092A>G, and c.1244–17T>A). Lastly, for 15 variants, we observed no effect on splicing and could only observe wild-type transcripts. Among those, eight variants were NCSS variants. Since there is no other mechanism that could explain potential pathogenicity other than splicing, we can reclassify these variants as benign by applying the BP7_Strong (RNA) rule (c.353+4A>T, c.725+9A>G, c.858+4A>G, c.1244–17T>A, c.1338+8A>G, c.1339–4A>G, c.1450+3A>G, and c.1450+20T>C).

## 4. Conclusions and Discussion

In this study, we showed that in vitro splice assays using minigenes can be used to assess splicing defects in the *RPE65* gene. We created a fast and efficient strategy to test for splicing defects with potential therapeutic implications. Even though variants that are listed in LOVD and ClinVar are not necessarily observed in a patient together with a second pathogenic allele in *RPE65*, these results are still valuable when these variants are detected in future patients.

Using these splice assays, we can reclassify seven variants to pathogenic, as there was no residual wild-type transcript observed. We cannot, however, reclassify any of the 13 variants that showed up to 20% of wild-type transcripts in our splicing assay. In the case of *RPE65*, when less than 5% of wild-type transcripts is present, a “downgraded” ACMG criterium can be given (PVS1-strong). For studies reporting isomerhydrolase activity, the cut-off for assigning the PS3-supporting criterium is ≤10% enzyme activity compared to a wild-type control (see ClinGen recommendations for *RPE65*). It is, however, explicitly advised against applying this rule for splicing assays. For variants with ≤10% of wild-type transcript, it could be recommended to test their effect on splicing in more organ-specific cell lines, such as iPSC-derived RPE cells, as it is known that some splice effects are tissue specific and with more precise quantification methods [11,12]. Furthermore, a system where the severity of the splicing defect of the variant of interest is taken together with the variant found on the second allele could be of use to improve the classification for *RPE65* variants. Although not completely comparable from a genotype–phenotype correlation point of view, this is used for the interpretation of *ABCA4* variants in patients with Stargardt disease and other forms of macular degeneration [13]. In this situation, *RPE65* variants that, for example, show <10% of wild-type transcripts in combination with a null-allele could be considered pathogenic, while this would not be the case when found in combination with another type of variant. This would also require studies that correlate the amount of residual wild-type transcript and enzymatic activity of RPE65, which is now lacking. Ultimately, this could lead to a refined, gene-specific threshold for residual wild-type transcript in splice assays to test *RPE65* splicing defects that could also be incorporated in the gene-specific ClinVar guidelines.

SpliceAI is widely used to predict effects on splicing, and while sample sizes are not sufficient to draw any conclusions on this, it is interesting to note that four out of the seven variants that are now considered pathogenic have a SpliceAI delta score between 0.4 and 0.5. This underlines the importance of testing the effect on splicing for variants with SpliceAI delta scores higher than 0.2 and not drawing any conclusions on the effect of the variant on splicing based solely on the SpliceAI score. This difference is also exemplified by the results of the construct harboring the c.1339–3C>G and c.1339–4A>G variants. While both variants had relatively high SpliceAI scores (0.84 and 0.81, respectively), the construct carrying the c.1339–3C>G variant showed complete absence of the wild-type transcript, while the c.1339–4A>G construct showed no aberrant splicing and only the wild-type transcript. This observation underlines that splice assays are key in confirming splicing defects, especially for NCSS variants, and that effects on splicing cannot be imputed from studies performed on a neighboring variant. Furthermore, in this study, we included variants with a SpliceAI score of >0.1 to assess effects on splicing in variants that would normally be excluded from analysis. In total, 30 tested variants were from this group of variants, but none of them had a pathogenic effect on splicing, as the most severe variants had 11% residual wild-type RNA (c.1244–17T>A). This suggests that for *RPE65* variants with a SpliceAI score between 0.1 and 0.2, testing using splice assays is not as relevant.

Here, we describe a comprehensive assay to assess splicing defects in *RPE65*. These findings contribute to the interpretation of variants found in patients that will in turn dictate their eligibility for gene therapy.

## Figures and Tables

**Figure 1 genes-16-01022-f001:**
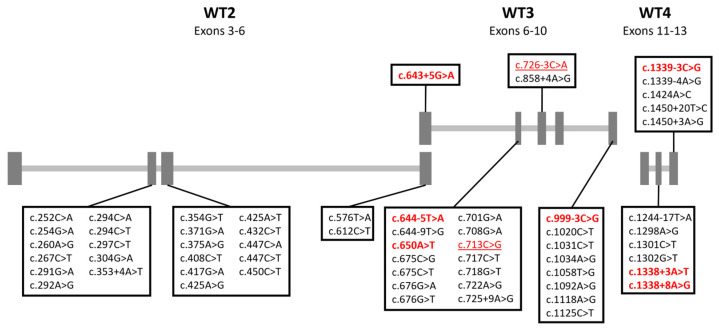
Overview of all wild-type constructs and the distribution of variants. Variants with no wild-type transcripts are in bold and red. Variants that have between 5% and 10% wild-type transcripts are underlined and in red. The WT-1 construct was omitted since no variants were tested using this construct.

**Table 1 genes-16-01022-t001:** Summary of included variants. Information on all variants that were tested in this study. Per variant, the highest SpliceAI Δ-score is highlighted in gray. ACMG, American College of Medical Genetics; B, Benign; LB, Likely benign; LOVD, Leiden Open Variant Database, LP, Likely pathogenic; NCSS, Non-canonical splice site; VUS, Variants of uncertain significance.

											Position to Predicted Gain/Loss			
c. Notation	p. Notation	Variant Type	ACMG Classification August 2023	ACMG Classification October 2024	Source	Wild-Type Construct	Acceptor Gain	Acceptor Loss	Donor Gain	Donor Loss	Acceptor Gain	Acceptor Loss	Donor Gain	Donor Loss
c.12G>T	p.(Gln4His)	Missense	VUS	VUS	ClinVar	WT-1	0.31	0.03	0.00	0.04	68	0	−1662	−82
c.252C>A	p.(Ile84=)	Synonymous	VUS	VUS	ClinVar	WT-2	0.00	0.25	0.00	0.18	−346	6	57	−101
c.254G>A	p.(Arg85His)	Missense	VUS	LP	ClinVar	WT-2	0.00	0.13	0.00	0.07	−344	8	8	−99
c.259G>A	p.(Asp87Asn)	Missense	VUS	VUS	LOVD	WT-2	0.00	0.25	0.00	0.19	−339	13	64	−94
c.260A>G	p.(Asp87Gly)	Missense	VUS	LP	ClinVar	WT-2	0.00	0.12	0.00	0.04	−338	14	1	−93
c.267C>T	p.(Tyr89=)	Synonymous	LB	LB	ClinVar	WT-2	0.00	0.15	0.00	0.10	−331	21	−4009	−86
c.268G>A	p.(Val90Ile)	Missense	VUS	VUS	ClinVar	WT-2	0.00	0.16	0.00	0.08	−330	22	73	−85
c.283G>C	p.(Glu95Gln)	Missense	VUS	LP	ClinVar	WT-2	0.00	0.14	0.00	0.06	−3	37	2014	−70
c.291G>A	p.(Arg97=)	Synonymous	LB	LB	ClinVar	WT-2	0.00	0.11	0.00	0.08	−307	45	−3985	−62
c.292A>G	p.(Ile98Val)	Missense	VUS	VUS	ClinVar	WT-2	0.00	0.27	0.00	0.25	−306	46	−3984	−61
c.294C>A	p.(Ile98=)	Synonymous	VUS	VUS	ClinVar	WT-2	0.00	0.37	0.00	0.46	1997	48	99	−59
c.294C>T	p.(Ile98=)	Synonymous	VUS	VUS	ClinVar	WT-2	0.00	0.27	0.00	0.25	−304	48	99	−59
c.297C>T	p.(Val99=)	Synonymous	LB	LB	ClinVar	WT-2	0.00	0.23	0.00	0.19	−301	51	102	−56
c.304G>A	p.(Glu102Lys)	Missense	VUS	LP	LOVD	WT-2	0.00	0.23	0.00	0.21	−294	58	2035	−49
c.353+4A>T	p.(?)	NCSS	VUS	VUS	ClinVar	WT-2	0.00	0.26	0.00	0.22	−162	111	−17	4
c.354G>T	p.(Arg118Ser)	Missense	VUS	LP	LOVD	WT-2	0.01	0.73	0.00	0.66	−20	0	−1480	−141
c.371G>A	p.(Arg124Gln)	Missense	VUS	VUS	ClinVar	WT-2	0.03	0.13	0.00	0.11	−3	17	−3	−124
c.375A>G	p.(Gly125=)	Synonymous	VUS	VUS	ClinVar	WT-2	0.00	0.24	0.00	0.20	−120	21	0	−120
c.408C>T	p.(Val136=)	Synonymous	LB	LB	ClinVar	WT-2	0.01	0.13	0.00	0.11	−8	54	1225	−87
c.417G>A	p.(Val139=)	Synonymous	VUS	VUS	ClinVar	WT-2	0.04	0.20	0.00	0.13	1	63	−2	−78
c.425A>G	p.(Asp142Gly)	Missense	VUS	VUS	LOVD	WT-2	0.00	0.21	0.00	0.16	−70	71	1	−70
c.425A>T	p.(Asp142Val)	Missense	VUS	VUS	ClinVar	WT-2	0.00	0.15	0.00	0.12	−70	71	1242	−70
c.447C>A	p.(Thr149=)	Synonymous	LB	LB	ClinVar	WT-2	0.00	0.14	0.00	0.11	−48	93	−967	−48
c.447C>T	p.(Thr149=)	Synonymous	LB	LB	ClinVar	WT-2	0.00	0.24	0.00	0.17	−26	93	−1387	−48
c.450C>T	p.(Asn150=)	Synonymous	LB	LB	ClinVar	WT-2	0.00	0.15	0.00	0.11	−45	96	1267	−45
c.494A>T	p.(Gln165Leu)	Missense	VUS	VUS	ClinVar	WT-2	0.00	0.58	0.21	0.83	1698	140	−18	−1
c.576T>A	p.(Ile192=)	Synonymous	VUS	LB	ClinVar	WT-2	0.14	0.14	0.11	0.15	−2	80	−1	−67
c.612C>T	p.(Tyr204=)	Synonymous	LB	LB	ClinVar	WT-2	0.02	0.11	0.02	0.12	404	116	35	−31
c.643+5G>A	p.(?)	NCSS	VUS	VUS	ClinVar	WT-3	0.00	0.42	0.06	0.42	5	152	71	5
c.644–14C>G	p.(?)	NCSS	VUS	VUS	ClinVar	WT-3	0.01	0.32	0.00	0.24	1632	−14	1197	−95
c.644–5T>A	p.(?)	NCSS	VUS	VUS	ClinVar	WT-3	0.02	0.42	0.01	0.33	1641	−5	1206	−86
c.644–9T>G	p.(?)	Missense	VUS	VUS	ClinVar	WT-3	0.00	0.18	0.00	0.18	1637	−9	1202	−90
c.650A>T	p.(Glu217Val)	Missense	VUS	VUS	ClinVar	WT-3	0.01	0.40	0.01	0.31	−1320	6	1217	−75
c.652G>T	p.(Asp218Tyr)	Missense	VUS	VUS	ClinVar	WT-3	0.01	0.31	0.00	0.23	1654	8	1219	−73
c.675C>G	p.(Ile225Met)	Missense	LB	B	LOVD	WT-3	0.00	0.17	0.00	0.14	−288	31	1242	−50
c.675C>T	p.(Ile225=)	Synonymous	VUS	VUS	ClinVar	WT-3	0.00	0.27	0.00	0.20	1677	31	1242	−50
c.676G>A	p.(Val226Ile)	Missense	VUS	VUS	LOVD	WT-3	0.00	0.25	0.00	0.18	1390	32	1243	−49
c.676G>T	p.(Val226Phe)	Missense	VUS	VUS	ClinVar	WT-3	0.00	0.29	0.00	0.22	1678	32	1243	−49
c.701G>A	p.(Arg234Gln)	Missense	VUS	VUS	ClinVar	WT-3	0.00	0.21	0.00	0.19	−25	57	1268	−24
c.708G>A	p.(Lys236=)	Synonymous	LB	VUS	ClinVar	WT-3	0.01	0.22	0.00	0.19	−18	64	−147	−17
c.713C>G	p.(Ser238Cys)	Missense	VUS	VUS	LOVD	WT-3	0.00	0.25	0.00	0.22	1715	69	1280	−12
c.717C>T	p.(Tyr239=)	Synonymous	LB	LB	ClinVar	WT-3	0.01	0.18	0.00	0.16	−9	73	1284	−8
c.718G>T	p.(Val240Phe)	Missense	VUS	LP	ClinVar	WT-3	0.01	0.14	0.00	0.10	−8	74	1285	−7
c.722A>G	p.(His241Arg)	Missense	VUS	LP	ClinVar	WT-3	0.00	0.07	0.03	0.12	1724	78	1	−3
c.725+4A>G	p.(?)	NCSS	VUS	VUS	LOVD	WT-3	0.01	0.39	0.00	0.33	1731	85	1655	4
c.725+6T>C	p.(?)	NCSS	VUS	VUS	ClinVar	WT-3	0.00	0.20	0.00	0.19	5	87	1657	6
c.725+9A>G	p.(?)	NCSS	LB	LB	ClinVar	WT-3	0.00	0.12	0.00	0.09	8	90	−121	9
c.726–3C>A	p.(?)	NCSS	VUS	VUS	LOVD	WT-3	0.07	0.81	0.00	0.36	100	−3	100	−135
c.858+4A>G	p.(?)	NCSS	VUS	VUS	ClinVar	WT-3	0.00	0.43	0.00	0.26	479	136	0	4
c.998G>A	p.(Gly333Glu)	Missense	VUS	LP	ClinVar	WT-3	0.01	0.31	0.16	0.39	700	−626	−155	0
c.999–3C>G	p.(?)	NCSS	VUS	VUS	ClinVar	WT-3	0.01	0.76	0.00	0.65	−188	−3	65	−132
c.1020C>T	p.(Tyr340=)	Synonymous	LB	LB	ClinVar	WT-3	0.12	0.15	0.00	0.09	−11	21	89	−108
c.1031C>T	p.(Ala344Val)	Missense	VUS	VUS	ClinVar	WT-3	0.00	0.12	0.00	0.09	−153	32	32	−97
c.1034A>G	p.(Asn345Ser)	Missense	VUS	VUS	ClinVar	WT-3	0.01	0.14	0.00	0.09	3	35	103	−94
c.1040G>T	p.(Arg347Leu)	Missense	VUS	LP	ClinVar	WT-3	0.00	0.10	0.00	0.11	−144	41	41	−88
c.1053A>G	p.(Glu351=)	Synonymous	VUS	VUS	ClinVar	WT-3	0.00	0.14	0.00	0.10	−131	54	14	−75
c.1058T>G	p.(Val353Gly)	Missense	VUS	VUS	ClinVar	WT-3	0.00	0.28	0.00	0.20	−126	59	824	−70
c.1092A>G	p.(Glu364=)	Synonymous	VUS	VUS	ClinVar	WT-3	0.01	0.33	0.22	0.41	−92	93	0	−36
c.1118A>G	p.(Asn373Ser)	Missense	VUS	VUS	ClinVar	WT-3	0.00	0.18	0.00	0.16	−66	119	187	−10
c.1125C>T	p.(Asp375=)	Synonymous	LB	LB	ClinVar	WT-3	0.00	0.11	0.01	0.08	−59	126	33	−3
c.1244–17T>A	p.(?)	NCSS	LB	LB	ClinVar	WT-4	0.00	0.15	0.00	0.15	485	−17	183	−111
c.1244–4G>A	p.(?)	NCSS	LB	LB	ClinVar	WT-4	0.00	0.08	0.00	0.11	−204	−4	91	−98
c.1269C>T	p.(Tyr423=)	Synonymous	LB	LB	ClinVar	WT-4	0.00	0.11	0.00	0.14	−2497	25	225	−69
c.1298A>G	p.(Tyr433Cys)	Missense	VUS	VUS	ClinVar	WT-4	0.00	0.10	0.01	0.11	−237	54	1	−40
c.1302G>T	p.(Ala434Tyr)	Missense	LB	LB	ClinVar	WT-4	0.00	0.12	0.00	0.12	−233	58	−1596	−36
c.1334A>G	p.(Asp445Gly)	Missense	VUS	LP	ClinVar	WT-4	0.67	0.00	0.93	0.44	90	−4	1	−4
c.1338+3A>T	p.(?)	NCSS	VUS	VUS	ClinVar	WT-4	0.02	0.24	0.01	0.47	599	97	524	3
c.1338+8A>G	p.(?)	NCSS	VUS	VUS	ClinVar	WT-4	0.01	0.10	0.00	0.15	−189	102	197	8
c.1339–3C>G	p.(?)	NCSS	VUS	VUS	LOVD	WT-4	0.79	0.84	0.00	0.34	−1	−3	−538	−114
c.1339–4A>G	p.(?)	NCSS	VUS	VUS	ClinVar	WT-4	0.81	0.12	0.00	0.06	−1	−4	−411	−115
c.1424A>C	p.(His475Pro)	Missense	VUS	VUS	ClinVar	WT-4	0.25	0.08	0.00	0.16	−6	85	−74	−26
c.1450+20T>C	p.(?)	NCSS	LB	LB	ClinVar	WT-4	0.04	0.09	0.01	0.13	40	−1118	−28	20
c.1450+3A>G	p.(?)	NCSS	VUS	VUS	ClinVar	WT-4	0.05	0.27	0.02	0.85	23	−1135	−45	3

**Table 2 genes-16-01022-t002:** Results of transcript quantification. Quantification was obtained using densitometry of gel electrophoresis after RT-PCR of the RNA of transfected cells. RNA and protein annotations were adapted according to quantification of transcripts and Sanger sequencing results. ACMG, American College of Medical Genetics; B, Benign; LB, Likely benign; LP, Likely pathogenic; NCSS, Non-canonical splice site; VUS, Variants of uncertain significance. * *p* < 0.05.

cDNA Variant	Protein Variant	ACMG Classification August 2023	ACMG Classification October 2024	Highest SpliceAI Score	Protein Variant	RNA Variant	Percentage of WT Transcript (%)	Percentage of Aberrant Transcript (%)	Identity of Aberrant Transcript(s)
c.643+5G>A	p.(?)	VUS	VUS	0.42	p.Val166Phefs*18	r.[496_725del]	0%	100%	Exon 6 and 7 skipping (83%), exon 6 skipping (8%), exon 7 skipping (3%), and unidentified transcript (6%)
c.644–5T>A	p.(?)	VUS	VUS	0.42	p.Asp215Valfs*4	r.[644_725del]	0%	100%	Exon 7 skipping
c.650A>T	p.(Glu217Val)	VUS	VUS	0.40	p.Asp215Valfs*4	r.[644_725del]	0%	100%	Exon 7 skipping
c.999–3C>G	p.(?)	VUS	VUS	0.76	p.[Phe334Leufs*7,Asp215Valfs*4]	r.[999_1128del,644_725del_999_1128del]	0%	100%	Exon 10 skipping (54%) and exon 7 + 10 skipping (46%)
c.1334A>G	p.(Asp445Gly)	VUS	LP	0.93	p.Asp445Alafs*3	r.[1333_1338del]	0%	100%	5 basepair 3′ truncation of exon 12 (87%) (13% is a PE identified as artefact)
c.1338+3A>T	p.(?)	VUS	VUS	0.47	p.Phe416Leufs*2	r.[1244_1338del]	0%	100%	Exon 12 skipping (3% artefact)
c.1339–3C>G	p.(?)	VUS	VUS	0.84	p.[Leu447Serfs*5,Phe416_Asp483del]	r.[1338_1339ins1339-2_1339-1,1244_1450del]	0%	100%	2 basepair 5′ elongation of exon 13 (34%), exon 12 + 13 skipping (41%), and exon 12 skipping (21%) (4% artefact)
c.713C>G	p.(Ser238Cys)	VUS	VUS	0.25	p.[Asp215Valfs*4,=]	r.[644_725del,=]	7%	93%	Exon 7 skipping (92%) and exon 7 + 10 skipping (1%)
c.726–3C>A	p.(?)	VUS	VUS	0.81	p.[Asp215Glyfs*7,=]	r.[644_858del,=]	9%	91%	Exon 7 and 8 skipping (88%) and intron 7 retention (3%)
c.1244–17T>A	p.(?)	LB	LB	0.15	p.[Phe416Leufs*2,=]	r.[1244_1338del,=]	11%	89%	Exon 12 skipping (87%) (1% artefact)
c.708G>A	p.(Lys236=)	LB	VUS	0.22	p.[Asp215Valfs*,=]	r.[644_725del,=]	12%	88%	Exon 7 skipping (87%) and exon 7 + 10 skipping (1%)
c.294C>A	p.(Ile98=)	VUS	VUS	0.46	p.[Phe83_Arg118delinsLeu,=]	r.[246_353del,=]	13%	87%	Exon 4 skipping (83%) and exon 4 + 5 skipping (4%)
c.676G>T	p.(Val226Phe)	VUS	VUS	0.29	p.[Asp215Valfs*4,=]	r.[644_725del,=]	13%	87%	Exon 7 skipping (86%) and exon 7 + 10 skipping (1%)
c.675C>T	p.(Ile225=)	VUS	VUS	0.27	p.[Asp215Valfs*4,=]	r.[644_725del,=]	14%	86%	Exon 7 skipping (85%) and exon 7 + 10 skipping (1%)
c.676G>A	p.(Val226Ile)	VUS	VUS	0.25	p.[Asp215Valfs*4,=]	r.[644_725del,=]	14%	86%	Exon 7 skipping (85%) and exon 7 + 10 skipping (1%)
c.717C>T	p.(Tyr239=)	LB	LB	0.18	p.[Asp215Valfs*4,=]	r.[644_725del,=]	15%	85%	Exon 7 skipping (80%) and exon 7 + 10 skipping (5%)
c.1092A>G	p.(Glu364=)	VUS	VUS	0.41	p.[Phe334Leufs*7,Asp215Valfs*4,=]	r.[999_1128del,644_725del_999_1128del,=]	16%	84%	Exon 10 skipping (38%), exon 7 + 10 skipping (43%), and exon 7 skipping (4%).
c.701G>A	p.(Arg234Gln)	VUS	VUS	0.21	p.[Asp215Valfs*4,Arg234Gln]	r.[644_725del,=]	19%	81%	Exon 7 skipping (76%), exon 7 + 10 skipping (5%), and an unidentified transcript (3%).
c.718G>T	p.(Val240Phe)	VUS	LP	0.14	p.[Asp215Valfs*4,Val240Phe]	r.[644_725del,=]	20%	80%	Exon 7 skipping (75%) and exon 7 + 10 skipping (5%)
c.1034A>G	p.(Asn345Ser)	VUS	VUS	0.14	p.[Asp215Valfs*4,Asn345Ser,Phe334Leufs*7]	r.[644_725del,=,644_725del_999_1128del,999_1128del]	23%	77%	Exon 7 skipping (43%), exon 10 skipping (16%), and exon 7 + 10 skipping (19%)
c.722A>G	p.(His241Arg)	VUS	LP	0.12	p.[Asp215Valfs*4,His241Arg]	r.[644_725del,=]	23%	77%	Exon 7 skipping (68%) and exon 7 + 10 skipping (9%)
c.1058T>G	p.(Val353Gly)	VUS	VUS	0.28	p.[Asp215Valfs*4,Val353Gly,Phe334Leufs*7]	r.[644_725del_999_1128del,644_725del,=,999_1128del]	24%	76%	Exon 7 skipping (22%), exon 10 skipping (15%), and exon 7 + 10 skipping (39%)
c.725+9A>G	p.(?)	LB	LB	0.12	p.[Asp215Valfs*4,=]	r.[644_725del,=]	26%	74%	Exon 7 skipping (65%) and exon 7 + 10 skipping (9%)
c.1302G>T	p.(Ala434Tyr)	LB	LB	0.12	p.[Phe416Leufs*2,=]	r.[1244_1338del,=]	26%	74%	Exon 12 skipping (73%) (1% artefact)
c.1125C>T	p.(Asp375=)	LB	LB	0.11	p.[Asp215Valfs*4,=,Phe334Leufs*7]	r.[644_725del,644_725del_999_1128del,=,999_1128del]	27%	73%	Exon 7 skipping (42%), exon 10 skipping (16%), and exon 7 + 10 skipping (15%)
c.354G>T	p.(Arg118Ser)	VUS	LP	0.73	p.[Arg118Ser,Phe119Leufs*6,Phe119Leufs*42]	r.[=,354_495del,353_354ins353+1_354-1]	28%	72%	Exon 5 skipping (37%), intron 4 retention (14%), exon 4 + 5 skipping (8%), and two unidentified transcripts (8% and 6%, resp.)
c.1031C>T	p.(Ala344Val)	VUS	VUS	0.12	p.[Asp215Valfs*4,Ala344Val]	r.[644_725del,=]	30%	70%	Exon 7 skipping (45%), exon 10 skipping (12%), and exon 7 + 10 skipping (13%)
c.1118A>G	p.(Asn373Ser)	VUS	VUS	0.18	p.[Asp215Valfs*4,Asn373Ser,Phe334Leufs*7]	r.[644_725del,644_725del_999_1128del,=,999_1128del]	32%	68%	Exon 7 skipping (35%), exon 10 skipping (16%), and exon 7 + 10 skipping (17%)
c.1020C>T	p.(Tyr340=)	LB	LB	0.15	p.[Asp215Valfs*4,=]	r.[644_725del,=]	33%	67%	Exon 7 skipping (50%), exon 10 skipping (10%), and exon 7 + 10 skipping (7%)
c.644–9T>G	p.(?)	VUS	VUS	0.18	p.[Asp215Valfs*4,=]	r.[644_725del,=]	36%	64%	Exon 7 skipping (62%) and exon 7 + 10 skipping (2%)
c.675C>G	p.(Ile225Met)	LB	B	0.17	p.[Asp215Valfs*4,Ile225Met]	r.[644_725del,=]	40%	60%	Exon 7 skipping (58%) and exon 7 + 10 skipping (2%)
c.858+4A>G	p.(?)	VUS	VUS	0.43	p.[=,Asp215Valfs*4]	r.[=,644_725del]	40%	60%	Exon 7 skipping (48%) and exon 7 + 8 skipping (12%)
c.1450+3A>G	p.(?)	VUS	VUS	0.85	p.[=,Val485_Ser533delinsAspGluSerAsnCysCysVal,Phe416_Asp483del]	r.[=,1450_1451ins1450+1_1450+48,1244_1450del]	49%	51%	Exon 13 elongation until 1450 + 48 (15%) and exon 12 + 13 skipping (37%)
c.1338+8A>G	p.(?)	VUS	VUS	0.15	p.[=,Phe416Leufs*2]	r.[=,1244_1338del]	55%	45%	Exon 12 skipping (44%) (1% artefact)
c.260A>G	p.(Asp87Gly)	VUS	LP	0.12	p.[Asp87Gly,Phe83_Arg118delinsLeu]	r.[=, 246_353del]	67%	33%	Exon 4 skipping (33%)
c.294C>T	p.(Ile98=)	VUS	VUS	0.27	p.[=,Phe83_Arg118delinsLeu]	r.[=,246_353del]	71%	29%	Exon 4 skipping (27%) and exon 4 + 5 skipping (2%)
c.292A>G	p.(Ile98Val)	VUS	VUS	0.27	p.[Ile98Val,Phe83_Arg118delinsLeu]	r.[=,246_353del]	75%	25%	Exon 4 skipping (25%)
c.447C>A	p.(Thr149=)	LB	LB	0.14	p.(=)	r.=	81%	19%	Exon 5 skipping (15%) and exon 4 + 5 skipping (4%).
c.447C>T	p.(Thr149=)	LB	LB	0.24	p.(=)	r.=	81%	19%	Exon 5 skipping (14%) and of exon 4 + 5 skipping (6%).
c.417G>A	p.(Val139=)	VUS	VUS	0.20	p.(=)	r.=	82%	18%	Exon 5 skipping (10%), exon 4 + 5 skipping (1%), and one unidentified transcript (6%)
c.297C>T	p.(Val99=)	LB	LB	0.23	p.(=)	r.=	88%	12%	Exon 4 skipping (11%) and exon 4 + 5 skipping (1%)
c.450C>T	p.(Asn150=)	LB	LB	0.15	p.(=)	r.=	91%	9%	Exon 5 skipping (8%) and exon 4 + 5 skipping (1%)
c.425A>G	p.(Asp142Gly)	VUS	VUS	0.21	p.[Asp142Gly]	r.=	94%	6%	Exon 5 skipping (6%).
c.612C>T	p.(Tyr204=)	LB	LB	0.12	p.(=)	r.=	97%	3%	Exon 3 + 4 + 5 skipping (3%)
c.252C>A	p.(Ile84=)	VUS	VUS	0.25	p.(=)	r.=	100%	0%	
c.254G>A	p.(Arg85His)	VUS	LP	0.13	p.(Arg85His)	r.=	100%	0%	
c.267C>T	p.(Tyr89=)	LB	LB	0.15	p.(=)	r.=	100%	0%	
c.291G>A	p.(Arg97=)	LB	LB	0.11	p.(=)	r.=	100%	0%	
c.304G>A	p.(Glu102Lys)	VUS	LP	0.23	p.(Glu102Lys)	r.=	100%	0%	
c.353+4A>T	p.(?)	VUS	VUS	0.26	p.(=)	r.=	100%	0%	
c.371G>A	p.(Arg124Gln)	VUS	VUS	0.13	p.(Arg124Gln)	r.=	100%	0%	
c.375A>G	p.(Gly125=)	VUS	VUS	0.24	p.(=)	r.=	100%	0%	
c.408C>T	p.(Val136=)	LB	LB	0.13	p.(=)	r.=	100%	0%	
c.425A>T	p.(Asp142Val)	VUS	VUS	0.15	p.(Asp142Val)	r.=	100%	0%	
c.576T>A	p.(Ile192=)	VUS	LB	0.15	p.(=)	r.=	100%	0%	
c.1298A>G	p.(Tyr433Cys)	VUS	VUS	0.11	p.(Tyr433Cys)	r.=	100%	0%	
c.1339–4A>G	p.(?)	VUS	VUS	0.81	p.(=)	r.=	100%	0%	
c.1424A>C	p.(His475Pro)	VUS	VUS	0.25	p.(His475Pro)	r.=	100%	0%	
c.1450+20T>C	p.(?)	LB	LB	0.13	p.(=)	r.=	100%	0%	

## Data Availability

All data are available from the corresponding author upon reasonable request.

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
