# Peer review of "Minigene Splice Assays Allow Pathogenicity Reclassification of RPE65 Variants of Uncertain Significance"

_genes, 2025, doi:10.3390/genes16091022_

Round 1

Reviewer 1 Report

Comments and Suggestions for Authors

Please refer to the enclosed pdf file.

Author Response

We would like to thank the reviewers for their thoughtful feedback and considerations. Below we have a point-by-point response to each to the comments.

Please expand figures/ tables legends.

We have expanded the legends by including appropriate abbreviations that were overlooked.

Clarify why a window has been applied 5000 upstream and 1000 downstream each variant and predicted exon skipping/intron retention events.

While the original publication of SpliceAI by Jaganathan et al. (2019) utilizes a window of 50bp by default it is well known that an effect of splicing can be at a larger distance than 50bp. Meanwhile the default splicing window is 500bp to ensure the capture of relevant predictions in its proximity. Moreover, the interplay and predictions of strength of an acceptor site and donor site of an exon essential. To ensure maximum coverage of the RPE65 region including its predicted promotor region we used 5,000 bp up- and downstream of the variant to capture all possible splice defects.

How do the authors assess pathogenicity for homozygous recessive alleles and compound heterozygous? Can co-occurrence of different alleles result in haploinsufficiency?

Currently, there are no ACMG criteria that allow for taking this into consideration. As we mention in line 146, this would be very useful for the interpretation of variants in general. We are not aware of any effect of co-occurrence of certain splice variants that result in haploinsufficiency and expect that a combination of two variants that affect splice merely result in a proportionate reduction of mRNA.

How are these variants distributed across populations? Do the least penetrant variants (the ones with higher RPE65 residual transcript) segregate with latitude or within specific ethnic groups?

We assessed the allele frequencies of these variants did not observe any meaningful differences. Based on this, we have no evidence that they segregate with certain ethnic groups or geographical location.

Are the tested variants all recessive? Is there any de novo or rare variant with unknown inheritance?

As far as we are aware, all variants in this study were in a recessive state. However, since the variants were exported from ClinVar and the LOVD, we cannot exclude this option.

The apparent ratio between full length/ spliced variants is assessed by semi-quantitative RT-PCR based densitometry. Have the authors estimated the proteomic levels of such variants? Can any of these variants lead to protein heterocomplex acting as dominant negatives?

We have not predicted these effects and are unaware of any prediction software enable to predict such events. So far only one variant has been described to cause a dominant negative effect (RPE65: c.1430A>G). None of the variants that we tested are near this variant.

Functional assessment of the RPE65 protein level could encompass delays in the 11-cis retinal chromophore regeneration, recovery of photoreceptor sensitivity after photobleaching, retinoid isomerase activity in vitro, scotopic ERG amplitudes and thickness of retinal layers in retinal organoids: Have the authors carried out any of the above?

Unfortunately, all variants were obtained from ClinVar and LOVD and were anonymized. Therefore, we are unaware of any testing of the parameters above since we did not have access to patient statuses.

The tables may be presented in a more reader-friendly way. I suggest reducing the size of the tables or mention them as supplementary files.

We apologize for the presentation of the manuscript, and we agree that the orientation of the tables was to be improved upon. The current lay-out is the result of the MDPI auto-formatting and will make sure that this is adapted before potential publication.

Reviewer 2 Report

Comments and Suggestions for Authors

Title: Minigene Splice Assays Allow Pathogenicity Reclassification of RPE65 Variants of Uncertain Significance

Authors: Panneman, D. M., Boonen, E. G. M., Corradi, Z., Cremers, F. P. M., and Roosing, S.

Summary: In this paper, the authors evaluated the effect on splicing predicted by SpliceAI for cases where genetic screening revealed variants of uncertain significance (VUS). They used minigene assays to assess the effect on splicing for all variants and selected 73 variants that were classified as either VUS or benign and were predicted to affect splicing by SpliceAI. They generated constructs containing the variants of interest using Gateway cloning and assessed the effects on splicing. They demonstrated that in vitro splice assays using minigenes can be used to access splicing defects in the RPE65 gene. Evidence from splice assays provided data to reclassify seven variants from VUS to pathogenic. Inherited mutations of RPE65 are the cause of inherited retinal dystrophies. There is an FDA approved gene therapy (Luxturna) that delivers a corrected version of RPE65, thus enabling the protein to be produced and correcting the defect. The methods present here will help to provide a more comprehensive interpretation of variants found in patients and will help to determine eligibility for gene therapy.

Revisions needed:

Page 3, Line 67 – Change SplicAI to SpliceAI

Page 3, Table 1 – fix Table 1 headings. Make the table landscape orientation if necessary.

Page 3, Line 72, 73, and 74. These lines belong together below the table.

Page 8, Table 2 – fix Table 2 headings. Make the table landscape orientation if necessary.

Page 13, Line 110, 111, 112, and 113. These lines belong together below the table.

Page 15, Figure 1 – consider changing the color for bold, orange constructs as this color is difficult to see when looking at table and is important data.

Page 17, Line 188 – change suggest to suggests.

Page 18, Line 209 – References should include all author names. They have been truncated using … and eliminated the complete author list on the papers.

Page 19, Line 250, 251 and 252. These lines belong together below the table.

Page 19, Supplemental Table 2 – The primer sequences should either be all capitalized or all lower case.

Page 23, Line 253 and 254. These lines belong together below the table.

Author Response

We would like to thank the reviewers for their thoughtful feedback and considerations. Below we have a point-by-point response to each to the comments.

Reviewer 2

Page 3, Line 67 – Change SplicAI to SpliceAI

Adapted

Page 3, Table 1 – fix Table 1 headings. Make the table landscape orientation if necessary.

We agree that the orientation of the tables can be improved upon. The current lay-out is the result of the MDPI auto-formatting and we will ensure that this is adapted before potential publication.

Page 3, Line 72, 73, and 74. These lines belong together below the table.

Adapted

Page 8, Table 2 – fix Table 2 headings. Make the table landscape orientation if necessary.

See comment above

Page 13, Line 110, 111, 112, and 113. These lines belong together below the table.

Adapted

Page 15, Figure 1 – consider changing the color for bold, orange constructs as this color is difficult to see when looking at table and is important data.

Adapted

Page 17, Line 188 – change suggest to suggests.

Adapted

Page 18, Line 209 – References should include all author names. They have been truncated using … and eliminated the complete author list on the papers.

Adapted.

Page 19, Line 250, 251 and 252. These lines belong together below the table.

Adapted

Page 19, Supplemental Table 2 – The primer sequences should either be all capitalized or all lower case.

Adapted

Page 23, Line 253 and 254. These lines belong together below the table.

Adapted